# Prophylactic antibiotic use for infective endocarditis: a systematic review and meta-analysis

Sue S H Lean,[1] Eric Jou [ORCID] ,[2] Jamie Sin Ying Ho [ORCID] ,[3] Ernest G L Jou[1]

SSHL and EJ contributed equally.
JSYH and EGLJ contributed equally.

SSHL and EJ are joint first authors.
JSYH and EGLJ are joint senior authors.

[1]Department of Dental Medicine, Wei Gong Memorial Hospital, Miaoli, Taiwan
[2]Queens' College, University of Cambridge, Cambridge, UK
[3]Department of Medicine, National University Health System, Singapore

**Correspondence to**
Dr Eric Jou; ej290@cam.ac.uk

## ABSTRACT

**Objectives** Infective endocarditis (IE) is a devastating disease with a 50% 1-year mortality rate. In recent years, medical authorities across the globe advised stricter criteria for antibiotic prophylaxis in patients with high risk of IE undergoing dental procedures. Whether such recommendations may increase the risk of IE in at-risk patients must be investigated.

**Design** Prospectively registered systematic review and meta-analysis.

**Data sources** Medline, Embase, Scopus and ClinicalTrials.gov were searched through 23 May 2022, together with an updated search on 5 August 2023.

**Eligibility criteria** All primary studies reporting IE within 3 months of dental procedures in adults >18 years of age were included, while conference abstracts, reviews, case reports and case series involving fewer than 10 cases were excluded.

**Data extraction and synthesis** All studies were assessed by two reviewers independently, and any discrepancies were further resolved through a third researcher.

**Results** Of the 3771 articles screened, 38 observational studies fit the inclusion criteria and were included in the study for subsequent analysis. Overall, 11% (95% CI 0.08 to 0.16, $I^2$=100%) of IE are associated with recent dental procedures. *Streptococcus viridans* accounted for 69% (95% CI 0.46 to 0.85) of IE in patients who had undergone recent dental procedures, compared with only 21% (95% CI 0.17 to 0.26) in controls (p=0.003). None of the high-risk patients developed IE across all studies where 100% of the patients were treated with prophylactic antibiotics, and IE patients are 12% more likely to have undergone recent dental manipulation compared with matched controls (95% CI 1.00 to 1.26, p=0.048).

**Conclusions** Although there is a lack of randomised control trials due to logistic difficulties in the literature on this topic, antibiotic prophylaxis are likely of benefit in reducing the incidence of IE in high-risk patients after dental procedures. Further well-designed high-quality case-control studies are required.

**Trial registration number** CRD42022326664.

## STRENGTHS AND LIMITATIONS OF THIS STUDY

⇒ A comprehensive systematic review and meta-analysis interrogating the complex question of the association between dental procedures, infective endocarditis and antibiotic prophylaxis through multiple angles.
⇒ The studies included in the present analysis originate from broad geographical locations spanning four continents including North America, Europe, Asia and Australia allowing accurate global representation.
⇒ No randomised-controlled trials were included in the present systematic review and meta-analysis due to the lack of such study type in the literature on this important topic.

be around 25% during in-hospital admissions and almost 50% at 1-year follow-up.[2] Symptoms of IE include fever, chills, rigours, headaches, dyspnoea, cough, myalgia, arthralgia and general fatigue, and in addition to antibiotics up to 50% of patients may require surgical treatment.[3]

Mechanistically, IE occurs due to injury to the heart lining or valve, allowing pathogens transiently travelling in the blood to lodge. Patients at elevated risk of IE include those with previous or active heart diseases which increases the susceptibility of pathogen colonisation on injured heart tissue, in accordance with the American Heart Association (AHA) and European Society of Cardiology (ESC) guidelines (online supplemental table 1).[4–7] Importantly, dental procedures are associated with bacteraemia,[8 9] and antibiotic prophylaxis have long been used for dental procedures across the world in at-risk patients to reduce IE incidence. Nevertheless, there has been uncertainties on the efficacy of antibiotic prophylaxis,[10] and in 2008 the UK National Institute for Health and Care Excellence (NICE) clinical guidelines were updated advising against their routine use in the prevention of IE in patients undergoing dental procedures. This is in contrast to most

## INTRODUCTION

Infective endocarditis (IE) is an infection of the inner linings or valves of the heart with a near 100% mortality rate before the advent of antibiotics.[1] Mortality remains high despite modern advances in medicine, estimated to

countries globally including the USA and many European countries where antibiotic prophylaxis remains in-use, although there is a tendency towards stricter criteria over the years.[7 11] The updated 2007 AHA and 2009 ESC guidelines indicated that antibiotic prophylaxis is no longer recommended for moderate-risk patients, and instead only for high-risk patients.[6 7] On one hand, withholding antibiotic prophylaxis where there are no foreseeable benefits is cost effective and can prevent overtreatment and side effects. However, given that IE carries significant morbidity and mortality, it remains vital to ascertain whether measures to reduce or completely withhold antibiotic prophylaxis may lead to a higher incidence of IE secondary to dental procedures.

Despite the long history of antibiotic prophylaxis in IE prevention, there remains to be no clear evidence to support or refute their use due to the lack of high-quality study or trials.[10 12 13] This is further complicated by most studies using bacteraemia rather than IE as an endpoint as highlighted in recent systematic reviews.[14] In this article, we perform a systematic review and meta-analysis of the literature exploring the aetiology and underlying microbial causes of dental procedure-related IE, followed by assessing the association between IE incidence and dental manipulation. Focus is placed on analysing the prevalence of recent preceding dental manipulation in patients diagnosed with IE in relation to prophylactic antibiotic use. Furthermore, studies reporting IE incidence and outcome in high-risk patients who had undergone recent dental procedures are also interrogated. Finally, based on our findings we comment on the rationale and efficacy of prophylactic antibiotic use in high-risk patients in the prevention of IE.

## MATERIALS AND METHODS

The systematic review and meta-analysis protocol was prospectively registered on PROSPERO (CRD42022326664). Patients or the public were not involved in the design, or conduct, or reporting, or dissemination plans of our research. The main clinical question interrogated in this systematic review and meta-analysis is to assess the potential association of dental procedures and IE, and secondarily whether prophylactic antibiotic use may be of benefit in IE prevention relating to dental procedures. Due to the lack of randomised-controlled trials in the literature, we attempted to answer this pressing question through using a multifaceted approach. First, we assessed the potential association between IE and recent dental procedures through the inclusion of studies where incidence of recent dental procedures were reported in IE patients. Additionally, assessments comparing the causative organisms of IE associated with recent dental procedures to general IE cases were also performed in studies where such data is reported. Furthermore, studies where patients at risk of IE were treated with prophylactic antibiotics for dental

procedures were also interrogated to assess for any association between antibiotic prophylaxis and IE prevention.

### Study selection

A systematic search of four databases namely Medline, Embase, Scopus and ClinicalTrials.gov was performed from the date of inception to 23 May 2022, with the search terms "dent*" OR "*dontic*" AND "endocarditis" (online supplemental table 2). References of included studies and relevant reviews were handsearched for additional articles. Inclusion criteria were primary studies involving adults, reporting IE within 3 months of dental procedures. The full inclusion criteria are shown in online supplemental table 3. Randomised controlled trials, cohort studies, case–control studies, cross-sectional studies and case series with 10 or more cases were included. Studies that reported dental sources, but not specifically dental procedures were excluded. Conference abstracts, reviews, case reports and case series involving fewer than 10 cases and non-English studies were excluded. Title and abstracts were screened by two reviewers independently, and full texts were subsequently reviewed for included studies and those with discrepancies. Conflicts were resolved by discussion and involvement of a third researcher. An updated search was performed on 5 August 2023 using the same search strategy, of which a total of four studies satisfied the eligibility criteria.[15–18] The four studies identified in the updated search are discussed in the discussion section and the findings compared with the meta-analysis from the original search.

### Data extraction

Data were extracted onto a standardised form by two researchers independently, which included the country of publication, study design, population, type of dental procedure, proportion of IE, organism identified on blood culture, use of prophylactic antibiotics and outcomes.

### Exposure and outcomes

In this study, we first investigated the causative organisms for IE and compared this between patients who had recent dental procedures to those who did not. IE was diagnosed according to the Duke criteria. Dental procedures included tooth extraction, fillings, scaling, endodontal and periodontal treatment, implant insertion and root canal treatment. Causative organisms were identified by microbiology investigations. We then analysed the proportion of patients with IE who underwent dental procedures within 3 months prior to the diagnosis of IE. The proportion of patients with exposure to recent dental procedures were compared between patients with IE and those without IE. To understand the risk of IE post dental procedure, the proportion of high-risk patients who developed IE within 3 months post dental procedure was analysed. High-risk patients were defined based on the AHA/ESC guidelines (online supplemental table 1).

## Quality assessment

The quality of the studies included in the systematic review was assessed using the Newcastle Ottawa Scale for cohort studies, case–control studies and case series.

## Statistical analysis

Statistical tests were performed using the meta and metafor packages on R (R Core Team (2021)), based on the guidelines outlined in the Cochrane Handbook. Meta-analysis of IE and dental procedure proportions was performed using the random-effects inverse variance model with logit transformation and DerSimonian and Laird method, and subgroup analysis was performed based on the region of publication, year of publication and proportion of patients with antibiotic prophylaxis. The type of bacterial organisms isolated from blood cultures were also analysed by meta-analysis of proportions as detailed above. Random-effects meta-analysis of binary outcomes was performed for case–control studies using the inverse variance model, where proportion of IE cases with exposure to dental procedures was compared with controls without IE. Sensitivity analysis using the random-effects generalised linear mixed models was also performed. Between-study heterogeneity was presented using the $I^2$ statistic. Publication bias was assessed by funnel plot and Egger's test for funnel plot asymmetry for analysis with more than 10 studies. A p<0.05 was considered statistically significant.

## RESULTS

A total of 3771 articles were identified after removal of duplicates, 3272 articles were excluded at the title and abstract screening stage as they were not relevant to the topic, and a final total of 38 articles satisfied the inclusion criteria and were included in this review (online supplemental figure 1). The risk of bias assessment is shown in online supplemental table 4.

## IE aetiology and causative organisms

The development of IE follows a two-step process whereby initial damage to the endocardium provides a nidus allowing pathogen settlement and colonisation following a period of transient bacteraemia or fungaemia. Studies indicate that bacteria are the most common pathogen class responsible for IE.[19 20] Common oral commensals or pathogens that cause IE include *Streptococcus viridans*[21] and those of the HACEK (Haemophilus, Aggregatibacter, Cardiobacterium, Eikenella and Kingella) group.[22]

In this systematic review, 26 studies reported the microbial species responsible for the IE cases (table 1). Pooled analysis revealed that *Staphylococcus* species accounted for 27% (95% CI 0.23 to 0.31, $I^2$=85%) of IE cases overall (online supplemental figure 2A), while *Streptococcus* species accounted for 36% (95% CI 0.29 to 0.44, $I^2$=94%) (online supplemental figure 2B). *S. viridans* and *Enterococcus* are responsible for 28% (95% CI 0.22 to 0.36, $I^2$=92%) (online supplemental figure 2C) and 9% (95%

CI 0.07 to 0.12, $I^2$=79%) (online supplemental figure 2D) of total IE cases, respectively. Overall, 13% (95% CI 0.10 to 0.16, $I^2$=83%) of all IE cases are due to HACEK and other organisms (online supplemental figure 2E), while in another 13% (95% CI 0.09 to 0.18, $I^2$=94%) (online supplemental figure 2F) the causative organism could not be identified.

Twenty of the 26 studies presented pooled data from all IE patients regardless of cause, while the other six reported exclusively the causative microorganism in IE patients who received recent dental procedures (table 1). The majority of the 20 studies with pooled data (12 out of 20; 60%) found *Staphylococcus* to be the most common cause of IE, ranging from 14% to 75% of patients. This is closely followed by *Streptococcus* species collectively being the dominant cause in 7 out of the 20 pooled studies (35%), ranging from 27.5% to 57%. Of the *Streptococcus* species found in these studies, *S. viridans* are the most common and are likely associated with the subgroup that received prior dental treatment. Other *Streptococcus* species identified include *Streptococcus bovis* and *Streptococcus pneumonia* which account for the remaining cases. Of the six studies where the causative organisms of IE were portrayed specifically in patients whom received prior dental procedures, *Streptococcus* species were the most common (table 1). *S. viridans* accounted for the majority, ranging from 23.9% and up to 100% of the total IE cases. HACEK and other organisms tend to be the next most common, while *Staphylococcus* is rare unlike in the general population. Subgroup meta-analysis comparing studies reporting microbial causes of IE exclusively in patients who had undergone recent dental procedures compared with general IE cases revealed important differences in the underlying cause of IE (table 1). In those with prior dental procedures, *S. viridans* are responsible for a significantly higher proportion at 69% (95% CI 0.46 to 0.85; $I^2$=63.7%) of cases, compared with 21% (95% CI 0.17 to 0.26; $I^2$=89.0%) in general IE patients (p=0.003). Conversely, those with recent dental procedures are less likely (p=0.002) to suffer from *Staphylococcus* IE at only 3% (95% CI 0.01 to 0.14, $I^2$=0%) compared with 29% in general IE patients (95% CI 0.25 to 0.34, $I^2$=85.9%).

## Proportion of IE cases associated with dental procedures

Thirty-three studies assessed the proportion of IE patients who had recently undergone dental procedures within 3 months (table 2), with cohort sizes ranging from 11 to 138 876 patients.[23 24] The most common dental procedures reported include tooth extraction, fillings, scaling and oral surgeries, although most studies did not specify (21 out of 33, 64%). The proportion of patients who had undergone recent dental procedures, which are considered to be the most likely cause of the IE in these patients, varied widely between studies (2.5%–49.9%). This amounted to an overall 11% (95% CI 0.08 to 0.16, $I^2$=100%) (online supplemental figure 3) incidence of IE patients who had received recent dental procedures. Subgroup analysis by continent including North America,

**Table 1** Results of the meta-analyses performed with subgroup analysis

| Outcome | No of studies | N | Proportion (95% CI) | $I^2$ value | P value |
|---|---|---|---|---|---|
| Dental procedures in IE patients | 28 | 154 582 | 0.11 (0.08 to 0.15) | 99.5% | – |
| Year of patient recruitment | | | | | 0.372 |
| Before 2007 | 19 | | 0.12 (0.09 to 0.17) | 80.1% | |
| After 2007 | 9 | | 0.09 (0.04 to 0.18) | 99.8% | |
| Continent of publication | | | | | 0.989 |
| Asia | 9 | | 0.10 (0.04 to 0.22) | 97.7% | |
| North America | 6 | | 0.12 (0.08 to 0.17) | 55.8% | |
| Europe | 12 | | 0.12 (0.07 to 0.19) | 99.2% | |
| Australia | 1 | | 0.11 (0.05 to 0.21) | – | |
| Streptococcus bacteraemia | 16 | 3994 | 0.36 (0.28 to 0.44) | 93.7% | – |
| Dental procedures proportion | | | | | 0.003 |
| All patients had dental procedures | 3 | | 0.61 (0.43 to 0.76) | 68.6% | |
| Not all patients had dental procedures | 13 | | 0.32 (0.25 to 0.39) | 93.7% | |
| *Streptococcus viridans* bacteraemia | 23 | 3363 | 0.30 (0.22 to 0.40) | 91.7% | – |
| Dental procedures proportion | | | | | 0.003 |
| All patients had dental procedures | 6 | | 0.69 (0.46 to 0.85) | 63.7% | |
| Not all patients had dental procedures | 17 | | 0.21 (0.17 to 0.26) | 89.0% | |
| Staphylococcus bacteraemia | 25 | 5215 | 0.23 (0.18 to 0.30) | 83.2% | – |
| Dental procedures proportion | | | | | 0.002 |
| All patients had dental procedures | 6 | | 0.03 (0.01 to 0.14) | 0% | |
| Not all patients had dental procedures | 19 | | 0.29 (0.25 to 0.34) | 85.9% | |
| Enterococcus bacteraemia | 21 | 4767 | 0.07 (0.05 to 0.11) | 76.6% | – |
| Dental procedures proportion | | | | | 0.355 |
| All patients had dental procedures | 4 | | 0.003 (0.00 to 0.79) | 0% | |
| Not all patients had dental procedures | 17 | | 0.08 (0.06 to 0.11) | 79.0% | |
| HACEK bacteraemia | 24 | 5163 | 0.12 (0.10 to 0.16) | 82.5% | – |
| Dental procedures proportion | | | | | 0.385 |
| All patients had dental procedures | 5 | | 0.05 (0.004 to 0.36) | 0.0% | |
| Not all patients had dental procedures | 19 | | 0.13 (0.10 to 0.16) | 85.8% | |
| Negative bacteraemia | 23 | 5062 | 0.11 (0.07 to 0.16) | 93.6% | – |
| Dental procedures proportion | | | | | 0.411 |
| All patients had dental procedures | 5 | | 0.06 (0.01 to 0.25) | 54.7% | |
| Not all patients had dental procedures | 18 | | 0.12 (0.08 to 0.18) | 94.9% | |
| IE in high-risk patients who underwent dental procedures | 4 | 413 | 0.00 (0.00 to 1.00) | 0% | – |

HACEK, Haemophilus, Aggregatibacter, Cardiobacterium, Eikenella and Kingella; IE, infective endocarditis.

Europe, Asia and Australia revealed no differences in the proportion of IE patients with recent dental procedures (p=0.99) (table 1). Meta-regression for percentage of antibiotic use (intercept coefficient –0.809; p=0.653) and year of publication (intercept coefficient –0.005; p=0.772) revealed no significant relationship with the proportion of IE patients who underwent recent dental procedures (online supplemental figure 4).

**Proportion of patients who received dental procedures in case–control studies**

We further analysed the included case–control studies to assess for incidence of recent dental treatment in IE patients compared with controls without IE, which amounted to a total of six eligible studies including both conventional case–control along with case-crossover studies. Four studies were case-crossover studies, where

**Table 2** Characteristics of included studies on patients with infective endocarditis

| Author, year, country | Study type | Population | Sample size | Type of dental procedure | Dental procedure (%) | Prophylactic abx (%) | Study duration | Duration between dental procedure and IE |
|---|---|---|---|---|---|---|---|---|
| Takeda 2005, Japan | Retrospective cohort | CHD with IE | 183 | Not specified | 21.00 | 18.40% | 1971–1998 | NR |
| Durack 1983, USA | Case series | Antibiotic prophylaxis failure, IE post dental procedures | 52 | Not specified | 92.00 | 100%; 12% had AHA recommended regimen | 1979 | 5 weeks (median) |
| Porat Ben-Amy 2009, Israel | Retrospective case-crossover | IE | 170 | Tooth extraction, periodontal treatment, root canal treatment, implant insertion, others | 12.00 | 50% eligible patients | 2003–2005 | 3 months |
| Hricak 2013, Slovakia | Prospective cohort | IE | 606 | Not specified | 13.20 | NR | 1984–2006 | NR |
| Carmona2003, Spain | Retrospective cohort | IE | 115 | Tooth extraction, fillings, scaling, professional cleaning | 5.20 | NR | 1997–2001 | 3 months |
| Delahaye2016, France | Prospective cohort | IE | 318 | Not specified | 2.50 | NR | 2005–2016 | 3 months |
| Strom 1998, USA | Population-based case–control study | Case—IE; control— no IE | 273 case 273 control | Dental hygiene care, filling periodontal treatment, restorative dentistry, extraction, endodontic treatment, treatment of tooth abscess, mouth or gingival surgery, other | 16.8 in cases; 14.3 in controls | Cases: 5.1%; Controls: 1.1% | 1988–1990 | 2 months |
| Aziz 2010, USA | Retrospective cohort | IE undergoing heart surgery | 50 | Not specified | 16.00 | NR | 2000–2003 | NR |
| Chu 2004, New Zealand | Retrospective cohort | IE | 65 | Not specified | 10.80 | NR | 1997–2002 | NR |
| Chen 2015, Taiwan | Population-based case–control study | IE | 713 | Tooth extraction, surgery, dental scaling, periodontal treatment, and endodontic treatment | Cases: 12.2; controls: 10.8 | 2.80% | 1999–2012 | 3 months |
| Tubiana2017, France | Population-based cohort and case–crossover study | Positioning or replacement of prosthetic heart valves, with IE | 138 876 | Invasive and non-invasive | 49.9; 26.0 of procedures were invasive | 50.1% eligible patinets | 2008–2014 | 3 months |

**Table 2** Continued

| Author, year, country | Study type | Population | Sample size | Type of dental procedure | Dental procedure (%) | Prophylactic abx (%) | Study duration | Duration between dental procedure and IE |
|---|---|---|---|---|---|---|---|---|
| Imperiale 1990, USA | Case–control study | Case—IE after dental procedure with high-risk cardiac lesion; control—no IE after dental procedure | 8 cases 24 controls | Cleaning, filing, extraction | NA | Cases: 12.5%; Controls 62.5% | 1980–1986 | 3 months |
| Sett 1993, Canada | Retrospective cohort | IE in procine bioprothesis patients | 56 | Not specified | NR | NR | 1975–1988 | NR |
| Cukingnan1983, USA | Retrospective cohort | Early valve replacement for IE | 42 | Not specified | 5.00 | NR | 1969–1983 | NR |
| Grover 1991, India | Retrospective cohort | IE | 19 | tooth extraction | 5.30 | 0% | 1982–1989 | NR |
| Mudhumitha 2018, India | Retrospective cohort | IE | 120 | Not specified | 3.30 | NR | 2010–2015 | 3 months |
| Lacassin1995, France | Case–control study | IE | 171 cases, 171 controls | Not specified | Cases: 22; controls: 19 | NR | 1990–1991 | 3 months |
| Duval 2006, France | Retrospective survey | General population | 2805 | Not specified | 0.70 | 26.30% | 1998 | 1 month |
| Chirillo 2016, Italy | Prospective cohort | IE | 677 | Not specified | 4.70 | 40.60% | 2007–2010 | 2 months |
| Krcmery 2018, Slovakia | Prospective cohort | IE | 180 | Not specified | 20.50 | NR | 1984–2017 | NR |
| Duval 2017, France | Case–control study | Case—IE caused by oral streptococci; Control—IE caused by nonoral pathogens | 73 cases, 192 controls | Not specified | 8.80 | NR | 2008–2013 | 3 months |
| Weinberger 1990, Israel | Retrospective cohort | IE patients with mitral valve prolapse | 19 | Not specified | 78.90 | 0% | 1970–1987 | 2 months |
| Martin 2007, UK | Case series | IE cases after dental procedure with successful litigation | 83 | Exodontia, scaling, endodontics and minor oral surgery | NA | 16.90% | 1983–2005 | Mean 9 days (range 2–22 days) |
| Santinga 1976, USA | Case series | IE and prosthetic heart valve | 11 | Not specified | 9.10 | NR | 1976 | NR |
| Siegman-Igra 2010, Israel | Prospective cohort | Culture-positive IE | 212 | Not specified | 8.00 | NR | 1995–1998; 2003–2005 | 3 months |

Continued

**Table 2** Continued

| Author, year, country | Study type | Population | Sample size | Type of dental procedure | Dental procedure (%) | Prophylactic abx (%) | Study duration | Duration between dental procedure and IE |
|---|---|---|---|---|---|---|---|---|
| Carmona 2002, Spain | Retrospective cohort | IE | 103 | Extraction, scaling, filling | 5.80 | 0% | 1997–1999 | NR |
| Luk 2014, Canada | Retrospective cohort | IE treated with surgical valve explant | 209 | Not specified | 13.90 | NR | 2001–2012 | NR |
| Kim 2019, Korea | Retrospective cohort | IE in cancer and non-cancer | 170 | Not specified | 4.10 | NR | 2011–2015 | NR |
| Smith 1976, UK | Retrospective cohort | IE | 78 | Not specified | 5.10 | NR | 1969–1972 | NR |
| Ballesta 2022, Spain | Retrospective cohort | IE | 101 | Not specified | 21.00 | 0% | 2000–2017 | NR |
| Dominguez 2016, Spain | Prospective cohort | IE | 1807 | Not specified | 7.00 | 38.50% | 2008–2013 | NR |
| Loupa 2004, Greece | Prospective cohort | IE | 101 | Not specified | 13.00 | NR | 1997–2000 | 3 months |
| Chen 2018, Taiwan | Case series | IE | 9120 | Dental cleaning, scaling and root planing, simple extraction, complicated extraction, odontectomy in both simple case and complicated case, and periodontal surgery | Cases: 2.7; controls: 2.7 | NR | 2004–2013 | 3 months |

CHC, congenital heart diseases; IE, infective endocarditis; NA, not applicable; NR, not reported.

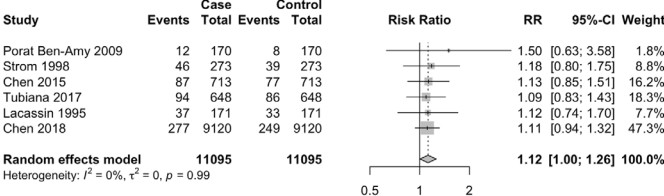

**Figure 1** Proportion of dental procedures in case versus control in case–control or cross-over studies. RR, relative risk.

cases were their own controls, thus minimising differences in baseline characteristics, while the other two studies used matched control populations to reduce confounding factors. Strom *et al* used age-matched, sex-matched and residence-matched controls,[25] while Lacassin *et al* matched the age, sex and underlying cardiac conditions between cases and controls.[26] Intriguingly, while none of the individual studies showed a statistically significant difference in the proportion of recent dental procedures comparing IE cases to controls, pooled meta-analysis revealed that IE patients are more likely to have received recent (within 3 months) dental treatment compared with matched controls (relative risk (RR) 1.12; 95% CI 1.00 to 1.26, $I^2$=0%, p=0.048) (figure 1). These findings support the notion that recent dental procedures may be associated with higher IE incidence.

## IE incidence in patients who underwent dental procedures

Five eligible studies investigated IE incidence in patients after receiving recent dental procedures (table 3). All of these are cohort studies, with three being retrospective while two prospective. One retrospective study with 12 102 patients with systemic lupus erythematosus from Taiwan did not mention the proportion of patients who received prophylactic antibiotics.[27] In that study, Cox multivariate proportional hazard analysis showed that recent dental procedures is an independent risk factor for IE (HR=36.8, p<0.001). The remaining four studies each reported that 100% of their patients received prophylactic antibiotics, consistent with their study design as all patients were at high risk of IE (figure 2), except in one study by Littner *et al* published in 1986 which also included moderate-risk patients treated with antibiotic prophylaxis as this was before the 2007 AHA and 2009 ESC guideline update.[28] In an Italian prospective cohort study of 104 patients with prosthetic heart valves, no patients developed IE 3 months after the dental surgery.[29] Another prospective cohort study (Israel) involving 90 patients with prosthetic heart valve or rheumatic heart disease similarly found no cases of IE after dental procedures (scaling, fillings, extraction, root canal, crowns and bridges).[30] In that study, the striking zero IE cases observed following an optimal protocol of administering

**Table 3** Characteristics of studies on patients undergoing dental procedures

| Author, year country | Study type | Population | Sample size | Type of dental procedure | IE (%) | Prophylactic antibiotics (%) | Study duration | Duration between dental procedure and IE |
|---|---|---|---|---|---|---|---|---|
| Russo 2000, Italy[29] | Prospective cohort | Prosthetic heart valve patients undergoing dental procedures | 104 | NR | 0 | 100 | NR | 3 months |
| Chang 2017, Taiwan[27] | Retrospective cohort | SLE patients | 12 102 | NR | NR | NR | NR | 30 days |
| Findler 2014, Israel[31] | Retrospective cohort | High risk for IE and underwent dental implant placement | 13 | All had dental implants | 0 | 100 | 1995–2012 | NA |
| Tzukert 1986, Israel[30] | Prospective cohort | Prosthetic heart valves, high-risk IE | 90 | Scaling, fillings, extraction, root canal, crowns, bridges | 0 | 100 | 1967–1982 | NA |
| Littner, 1986, Israel[28] | Retrospective cohort | Susceptible patients to IE | 206 | Extraction, drainage of abscess, biopsy, filling, oral surgery, crown, bridge, scaling | 0 | 100 | 1982–1986 | NA |

IE, infective endocarditis; NA, not applicable; NR, not reported; SLE, systemic lupus erythematosus.

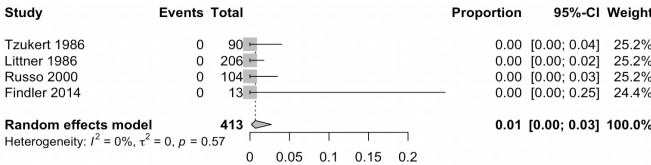

| Study | Events | Total | Proportion | 95%-CI | Weight |
|---|---|---|---|---|---|
| Tzukert 1986 | 0 | 90 | 0.00 | [0.00; 0.04] | 25.2% |
| Littner 1986 | 0 | 206 | 0.00 | [0.00; 0.02] | 25.2% |
| Russo 2000 | 0 | 104 | 0.00 | [0.00; 0.03] | 25.2% |
| Findler 2014 | 0 | 13 | 0.00 | [0.00; 0.25] | 24.4% |
| **Random effects model** | | **413** | **0.01** | **[0.00; 0.03]** | **100.0%** |

Heterogeneity: $I^2 = 0\%$, $\tau^2 = 0$, $p = 0.57$

0    0.05   0.1   0.15   0.2

**Figure 2** Proportion of IE in moderate-risk and high-risk patients who underwent dental procedures. IE, infective endocarditis.

appropriate prophylactic antibiotics proven to be efficacious against *S. viridans*, is significantly lower than what would be expected statistically. This suggests that appropriate antibiotic prophylaxis may reduce IE incidence. Similarly, the remaining two retrospective cohort studies both found no cases of IE following dental procedures in high-risk patients.[28 31] In one study, 13 patients received dental implants with no cases of IE at 2-year follow-up.[31] In the Littner *et al* study where 206 patients were assessed which included both moderate risk in addition to high-risk patients, none of them developed endocarditis. Of note, a minority (13.1%) developed mild side effects from prophylactic amoxicillin administration for dental procedures (extraction, abscess drainage, oral surgery, root canal, scaling, crowns and bridges).[28] In that study, the most common side effects were gastrointestinal, including abdominal discomfort (11 patients; 5.3%), nausea (6 patients; 2.9%) and constipation (5 patients; 2.4%). Overall, the severity of these side effects was classified as very mild, and patients showed a high degree of compliance to the prophylactic treatment. Collectively, these studies suggest that while recent dental procedures are independently associated with IE risk, prophylactic antibiotic use may be beneficial in reducing IE incidence.

## Publication bias

For the outcome of proportion of IE patients who underwent recent dental procedures, the funnel plot showed asymmetry, confirmed on Egger's test (t=−4.46, p=0.0001) (online supplemental figure 5). The meta-analysis of the proportion of Staphylococcus and Enterococcus IE showed asymmetrical funnel plots while the proportions of Streptococcus, *S. viridans*, HACEK and culture negative IE did not show asymmetrical funnel plots (online supplemental figure 5). Therefore, there may potentially be publication bias and small study effects, and further research is needed to confirm the findings of this study.

## DISCUSSION

In this systematic review and meta-analysis, we address the critical question of whether current antibiotic prophylaxis regimes are effective and of benefit in reducing IE incidence in high-risk patients. This is particularly crucial in an era where IE incidence continues to rise globally.[32]

We found *Streptococcus* species to be the most common cause of IE followed by *Staphylococcus* on pooled analysis of 28 eligible studies with suitable data. This differs from the current consensus where modern studies found *Staphylococcus* to be the leading cause of IE in the general population.[33] The overrepresentation of *Streptococcus* IE in our systematic review is likely a reflection of the included studies typically involving dental procedures. Accordingly, our meta-analysis found that microbial causes of IE is significantly different in IE patients who had undergone recent dental procedures compared with general IE cases, with a striking almost 70% prevalence of *S. viridans* IE in the former group. Conversely, we found that patients who received prior dental manipulation are much less likely to suffer from *Staphylococcus* IE at only 3% compared with 29% of the general IE patients. Other studies have similar found *Staphylococcus aureus* to account for approximately 30% of overall IE cases.[34] The association of *S. viridans* with post dental procedure IE observed here is consistent with the former being a common cause of odontogenic diseases.[35] These findings suggest that any prophylactic antibiotic use for dental procedures should be tailored to best protect against *S. viridans* to ensure maximal efficacy. Importantly, while amoxicillin is commonly used for antibiotic prophylaxis prior to dental procedures in at-risk patients, studies have shown *S. viridans* to be resistant to amoxicillin to differing degrees, and may contribute to the lack of efficacy of prophylactic antibiotics found in some studies.[36 37]

Gram negative organisms, in particular the oral commensals of the HACEK group, remain as important causes of IE despite accounting for only a small proportion of cases compared with gram positive cocci *Streptococcus* and *Staphylococcus*.[22] Our systematic review showed that many studies in the literature do not report HACEK organisms as a separate categorical cause of IE to other miscellaneous pathogens. Future studies segregating HACEK organisms from other bacterial causes of IE will shed insight on whether prophylactic antibiotics use is efficacious in preventing HACEK IE specifically.

We found that segregating studies into two groups defined as above or below average use of prophylactic antibiotics for dental procedures revealed that increased prophylactic antibiotic use may reduce IE from 18% to 11.5% although this was not statistically significant (p=0.49). Our findings cannot rule out antibiotics as important prophylactic treatment in reducing IE incidence in high-risk patients, as only a total of 11 eligible studies had suitable data for analysis, and as such, our meta-analysis is likely underpowered. Furthermore, most studies did not report the proportion of patients who are deemed high-risk in their respective cohorts, and therefore, studies with low prophylactic antibiotic usage may indeed have covered for all high-risk patients. In fact, our meta-analysis on all eligible studies investigating IE incidence after dental treatment with appropriate antibiotic cover in 100% of patients revealed a striking 0% incidence of IE. More studies are required to increase the sample size which will allow future meta-analyses to reach sufficient power to draw further conclusions.

Our meta-analysis of the six eligible studies with suitable data eliciting a pooled total of 11 095 cases and the same number of matched controls showed that IE patients are significantly more likely to have undergone recent dental procedures (RR 1.12, 95% CI 1.00 to 1.26, $I^2$=0%). We have selected 3 months as the cut-off for recent dental procedures due to the relatively long incubation period and prolonged duration of symptoms of IE, in accordance with the majority of previously published studies (table 2). These findings support the notion that dental procedures may increase the risk of IE and argue for antibiotic prophylaxis. As many of these studies utilised a case-crossover design (self-controls), this is unlikely to be due to other existing comorbidities. Indeed, studies assessing the incidence of IE before and after the 2007 AHA guideline changes found the sharp decline in prophylactic antibiotic use to be associated with a significant increase in IE incidence post dental procedures.[38] Dental procedures such as tooth extraction have been shown to cause bacteraemia in 100% of adults.[8 9] Studies in the UK interrogating the incidence of IE after the 2008 NICE guideline changes which advised against routine prophylactic antibiotic use for dental procedures in at-risk patients similarly found an increase in incidence of oral streptococcus IE but not overall IE.[39] This is consistent with our findings where oral streptococcus, mainly consisting of *S. viridans* group, are the major cause of IE after dental procedures whereas *Staphylococcus* species are the more common cause in general IE patients.

Recently, four important studies investigating IE, dental procedures and antibiotic prophylaxis were published[15–18] after the inclusion deadline of the present systematic review and meta-analysis as per prospectively registered on PROSPERO. In a US case-crossover study of 7 951 972 patients with employer-provided health cover, a strong temporal association was found between invasive dental procedures in the preceding 4 weeks and IE in high-risk patients, with an OR of 2.00 (95% CI 1.59 to 2.52, p=0.002).[16] The association was strongest with certain dental procedures including dental extractions (OR 11.08, 95% CI 7.34 to 16.74, p<0.0001) and oral-surgical procedures (OR 50.77, 95% CI 20.79 to 123.98, p<0.0001). Strikingly, in that study the author also found that prophylactic antibiotic administration reduced IE incidence after dental procedures (OR 0.49, 95% CI 0.29 to 0.85, p=0.01) including dental extractions (OR 0.13, 95% CI 0.03 to 0.34, p<0.0001) and oral-surgical procedures (OR 0.09, 95% CI 0.01 to 0.35, p=0.002), particularly in high-risk as opposed to moderate or low-risk patients.[16] In another elegant case-crossover study of 1 678 190 Medicaid patients (basic medical and dental cover), similar findings were observed where recent dental extractions (OR 14.17, 95% CI 5.40 to 52.11, p<0.0001) or oral surgery (OR 29.98, 95% CI 9.62 to 119.34, p<0.0001) within 30 days is associated with increased IE incidence, and that prophylactic antibiotic administration for dental procedure reduced IE incidence (OR 0.20, 95% CI 0.06 to 0.53, p<0.0001).[17] The number needed to treat with

antibiotic prophylaxis to prevent one case of IE was estimated to range between 71 and 244 depending on the type of dental procedure. Finally, in a UK-based study of 14 731 patients by the same group investigating various different procedures and IE, dental procedures including extraction and surgical tooth removal were found to be associated with increased IE risk (OR 2.14, 95% CI 1.22 to 3.76, p=0.047).[15] In a South Korean study of 62 019 patients with cardiac implantable electronic devices, invasive dental procedures were found to be associated with increased risk of IE (OR 1.75, 95% CI 1.48 to 2.05, p<0.001), with a mean time interval of around 60 days between receiving dental procedures and subsequently developing IE.[18] Overall, these findings are largely consistent with our systematic review and meta-analysis, where IE patients are associated with increased risk of having undergone recent dental procedures in the preceding 3 months (RR 1.12; 95% CI 1.00 to 1.26, $I^2$=0%, p=0.048), and that across all the studies where 100% of the patients received antibiotic prophylaxis, none of the high-risk patients developed IE. Importantly, the four new studies were not included in the results of our systematic review and meta-analysis due to being published after the pre-established search date as registered on PROSPERO. This is ideal as these studies have a very large sample size, one of which included 7 951 972 patients which is larger than all previous studies published in the last 50 years included in our meta-analysis combined. The striking concordance of our meta-analysis of all previous evidence to the recent large cohort studies indicates that there is an association between dental procedures and IE, and support the aforementioned AHA and ESC guidelines advocating the use of antibiotic prophylaxis in high-risk patients for dental procedures.

### Limitations

While our systematic review and meta-analysis provides an important appraisal and synthesis of all the available evidence on the association of dental procedures and IE in the context of prophylactic antibiotic use, there are important limitations. These include the differing patient characteristics between studies which may contribute to interstudy heterogeneity. Due to the update in AHA and ESC guidelines in 2007 and 2009, respectively, the use of prophylactic antibiotics may differ between studies performed in different years, which may confound the association of invasive dental procedures and IE. Further limitations include the lack of randomised controlled trials available in the literature, and therefore, studies included in this systematic review and meta-analysis mainly consist of observational cohort studies and surveys. In the comparison of IE patients with controls, case–control studies are susceptible to confounding factors, thus causality between dental procedures and IE cannot be definitively established. Due to the rarity of IE, a large sample size is required to allow sufficient statistical power to detect any potential association, if any, between prophylactic antibiotic use for recent dental procedures

and IE incidence. Indeed, while none of the individual included studies found a statistically significant difference in the incidence of recent dental procedures in IE patients compared with controls likely due to being underpowered, our meta-analysis demonstrated that IE patients are more likely to have undergone recent dental manipulation. The efficacy of prophylactic antibiotics, shown by some studies to be 100% in our meta-analysis, may further hide the association between IE and recent dental procedures. Another potential limitation is that studies performed in different times may record the casual organism differently, due to changes in classification of some types of bacteria, particularly enterococci and HACEK organisms. Furthermore, while some studies directly assessed dental records when gathering data on dental procedures, others relied on self-report by patients which is prone to recall bias. Similarly for data on prophylactic antibiotic use, some studies utilised patient records while others may rely on patient recollection or simply assume that all high-risk patients would have been offered antibiotic prophylaxis, therefore may be difficult to accurately verify.

## CONCLUSIONS

Overall, our findings in this systematic review and meta-analysis indicate that prophylactic antibiotics for dental procedures may be of benefit in preventing IE in high-risk patients. Given the logistic difficulties of conducting randomised control trials for prophylactic antibiotic use and IE, further well-designed cohort studies will be invaluable to address this very important issue.

**Contributors** SSHL, EJ, JSYH and EGLJ were involved in the conceptualisation of the study, designed the study, collected and analysed the data. All authors contributed to writing the manuscript and approved for the final version of the paper to be published. EJ is responsible for the overall content as the guarantor.

**Funding** The authors have not declared a specific grant for this research from any funding agency in the public, commercial or not-for-profit sectors.

**Competing interests** None declared.

**Patient and public involvement** Patients and/or the public were not involved in the design, or conduct, or reporting, or dissemination plans of this research.

**Patient consent for publication** Not applicable.

**Provenance and peer review** Not commissioned; externally peer reviewed.

**Data availability statement** All data relevant to the study are included in the article or uploaded as online supplemental information. All data relevant to the study are included in the article or uploaded as online supplemental information. No additional data available.

**ORCID iDs**
Eric Jou http://orcid.org/0000-0002-6259-4874
Jamie Sin Ying Ho http://orcid.org/0000-0002-1180-6902

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
