## [Reviewer comments · BMJ Open]

ARTICLE DETAILS

TITLE (PROVISIONAL)	Prophylactic antibiotic use for infective endocarditis - a systematic review and meta-analysis
AUTHORS	Lean, Sue SH; Jou, Eric; Ho, Jamie; Jou, Ernest GL

VERSION 1 – REVIEW

REVIEWER	Thornhill, M. University of Sheffield, School of Clinical Dentistry Other than being a researcher with expertise and publications related to this subject, I have no competing interests
REVIEW RETURNED	28-Jun-2023

GENERAL COMMENTS	This paper has improved significantly following the changes made by the authors in response to the reviewers comments. In particular the Limitations section now more accurately describes the main limitations of the study. Also the mention in the discussion of the 3 important new studies on this subject that appeared after the end date for the systematic review and before submission, has helped ensure the review is not too out of date. The discussion of these papers is appropriate and it is made clear they were published after the inclusion deadline of the present study. For clarity, it would be helpful to also state that because of this these 3 studies were not included in any of the analyses. or results. The issue of individuals at high-risk for IE is critical to all aspects of this paper. However, there is still not clarity about what constitutes high risk. Although the authors state (in their response to the reviewers) that they used the AHA/ESC definitions, this its still not clear in the manuscript. The only mention of what constitutes high-risk is in line 60-63 of the Introduction - "Patients at high risk of IE include those with previous or active heart disease which increases the susceptibility of pathogen colonisation on injured heart tissue, in accordance with the AHA and ESC guidelines". Strangely, the reference given here is for a summary of the NICE guidelines (ref 4). However, this definition is wide enough to include all those previously defined as moderate risk by the AHA and ESC (e.g. those with native valve disease (e.g. mitral valve prolapse), patients with congenital valve defects (e.g. bicuspid aortic valve), hypertrophic cardiomyopathy, rheumatic heart disease etc) as well as those defined as high risk (those with prosthetic or repaired heart valves, previous history of IE, or cyanotic congenital heart disease). The distinction between moderate and high risk is however critical. Prior to 2007 the AHA (and 2009 the ESC) recommended those at moderate as well as those at high risk received AP. After these date they recommended only those at high risk received AP. In the UK
--

	both groups were recommended for AP until NICE recommended against AP use for both groups in 2008. Moreover, recent studies have shown that any association between invasive dental procedures and IE, while increased in both groups, is much larger in those at high-risk. Furthermore, recent studies suggest that AP is much more effective in reducing IE risk following invasive dental procedures in those at high risk than those at moderate risk. Thus understanding the difference between these two groups is critical in understanding any association studies, the impact of changing guidelines and the likelihood that AP use will hide any association between the risk groups and subsequent IE. A clearer explanation (with examples) needs to be provided of what constitutes moderate- and high-risk patients (using the AHA/ESC definitions). This might most easily be done with a small table. Finally, in the main conclusion, the authors describe why randomised controlled trials are logistically difficult (if not impossible) in studying the efficacy of AP, and conclude therefore the further well-designed cohort studies will be invaluable to address this very important issue. However, in the abstract conclusions, they conclude "Further well-designed randomised controlled trial and high-quality case-controlled studies are required." These two conclusions seem in conflict with each other with regard to RCTs.
--	--

REVIEWER	Nabil, Syed Universiti Kebangsaan Malaysia, Oral and maxillofacial surgery
REVIEW RETURNED	02-Jul-2023

GENERAL COMMENTS	This paper has been reviewed previously and was commented on some of its limitation. Most of the previously highlighted comments has been addressed appropriately. In a few other comments the authors did acknowledge these issues and discussed it in the paper well. I have no further suggestion for improvement for this revised manuscript. It has been significantly improved and ready for publication in my view.
--

REVIEWER	Noma, Hisashi The Institute of Statistical Mathematics, Department of Data Science
REVIEW RETURNED	19-Jul-2023

GENERAL COMMENTS	 1. Explanations of the methods are insufficient. What were the definitions of individual outcomes and effect measures? They should be clearly explained, at least, corresponding to the Results section. 2. The statistical analysis methods should also be explained in detail, at least, corresponding to the Results section. 3. What study designs were considered in the study search? It should be explicitly mentioned. (study designs should be concretely mentioned) 4. How the risk of bias was evaluated? 5. How the proportion-type outcomes were defined? Several study designs were mixed, and I guess the estimation might be difficult for the case-control-type studies (especially if matching was used). The definitions should be clearly shown for individual study designs. 6. This study is a systematic review based on non-experimental studies. This is a relevant information, thus should be noted in Abstract. 7. The statistical software and packages used in data analyses should be noted. 8. What methods were used in the meta-analyses of proportion-type
--

	outcomes? (logit-transformation?) They should be accurately explained. 9. The synthesis result in Figure 2 is quite unnatural. The confidence interval was [0, 1], although those of individual studies were much narrower. Why such result was obtained under tau=0? I think this result is incorrect. 10. This study is a systematic review of non-experimental studies, so confounding is a relevant concern. The authors should mention for the control of confounding factors in individual studies, and should discuss how it potentially influences to the overall result. 11. The I² statistics were quite large for many outcomes, even for subgroup analyses. It might be caused by the differences of background factors of individual studies. I recommend to perform meta-regression analyses for primary outcomes and relevant factors. Also, the causes of heterogeneity should be discussed in the Discussion section. 12. Publication bias evaluations are recommended for primary outcomes.
--	---

VERSION 1 – AUTHOR RESPONSE

Reviewer: 1

This paper has improved significantly following the changes made by the authors in response to the reviewers comments. In particular the Limitations section now more accurately describes the main limitations of the study. Also the mention in the discussion of the 3 important new studies on this subject that appeared after the end date for the systematic review and before submission, has helped ensure the review is not too out of date. The discussion of these papers is appropriate and it is made clear they were published after the inclusion deadline of the present study. For clarity, it would be helpful to also state that because of this these 3 studies were not included in any of the analyses. or results.

We thank Reviewer 1 for the very insightful suggestions in the previous round of review and we agree with their comments that the paper has now “improved significantly following the changes”. The incorporation of the 3 new studies published after the previously established search date as per our PROSPERO protocol for this systematic review in the discussion section was an excellent suggestion by Reviewer 1 in the previous revision, and strengthens the findings of our study with the overall conclusions being highly concordant. Importantly, the three new studies (and an additional one identified in our updated search) have a significantly larger sample size (one of which with 7,951, 972 patients) compared to all the previous studies published (in the past circa 50 years) and included in our systematic review, and therefore the current analysis and arrangement of the manuscript is ideal as we were able to show that all of the previous studies published prior to the 4 new studies showed concordant results with the 4 new studies. Furthermore, had we included the new study with the almost 8 million patients in the meta-analysis, this would largely skew the results towards the conclusion of that very elegant study due to the very large sample size compared to all previous studies in the past circa 50 years combined. We have added the above rationale to the discussion section of the updated manuscript, highlighted in yellow.

The issue of individuals at high-risk for IE is critical to all aspects of this paper. However, there

is still not clarity about what constitutes high risk. Although the authors state (in their response to the reviewers) that they used the AHA/ESC definitions, this its still not clear in the manuscript. The only mention of what constitutes high-risk is in line 60-63 of the Introduction - "Patients at high risk of IE include those with previous or active heart disease which increases the susceptibility of pathogen colonisation on injured heart tissue, in accordance with the AHA and ESC guidelines". Strangely, the reference given here is for a summary of the NICE guidelines (ref 4). However, this definition is wide enough to include all those previously defined as moderate risk by the AHA and ESC (e.g. those with native valve disease (e.g. mitral valve prolapse), patients with congenital valve defects (e.g. bicuspid aortic valve), hypertrophic cardiomyopathy, rheumatic heart disease etc) as well as those defined as high risk (those with prosthetic or repaired heart valves, previous history of IE, or cyanotic congenital heart disease). The distinction between moderate and high risk is however critical. Prior to 2007 the AHA (and 2009 the ESC) recommended those at moderate as well as those at high risk received AP. After these date they recommended only those at high risk received AP. In the UK both groups were recommended for AP until NICE recommended against AP use for both groups in 2008. Moreover, recent studies have shown that any association between invasive dental procedures and IE, while increased in both groups, is much larger in those at high-risk. Furthermore, recent studies suggest that AP is much more effective in reducing IE risk following invasive dental procedures in those at high risk than those at moderate risk. Thus understanding the difference between these two groups is critical in understanding any association studies, the impact of changing guidelines and the likelihood that AP use will hide any association between the risk groups and subsiquent IE. A clearer explanation (with examples) needs to be provided of what constitutes moderate- and high-risk patients (using the AHA/ESC definitions). This might most easily be done with a small table.

We thank reviewer 1 of the excellent suggestion to include a table to summarise high and moderate risk IE patient features as per AHA/ESC guidelines, and have depicted this in a newly included Supplementary Table 1 in the updated manuscript. Additional text (highlighted in yellow) has also been added to the introduction section for further clarity on this. Updated references of the AHA/ESC guidelines have also been included.

Finally, in the main conclusion, the authors describe why randomised controlled trials are logistically difficult (if not impossible) in studying the efficacy of AP, and conclude therefore the further well-designed cohort studies will be invaluable to address this very important issue. However, in the abstract conclusions, they conclude "Further well-designed randomised controlled trial and high-quality case-controlled studies are required." These two conclusions seem in conflict with each other with regard to RCTs.

We thank Reviewer 1 for the comment and have updated the conclusions section of the abstract to illustrate the logistic difficulties of conducting RCTs and the importance of high-quality case-controlled studies.

Overall, we the authors wish to thank Reviewer 1 for the very constructive and thorough comments in the two rounds of revision, which we believe have significantly improved the manuscript.

Reviewer: 2

This paper has been reviewed previously and was commented on some of its limitation. Most of the previously highlighted comments has been addressed appropriately. In a few other comments the authors did acknowledge these issues and discussed it in the paper well. I have no further suggestion for improvement for this revised manuscript. It has been significantly improved and ready for publication in my view.

We thank Reviewer 2 for commenting that our paper has “been significantly improved and ready for publication”, and for the very helpful comments and constructive suggestions in the previous revisions.

Reviewer: 3

1. Explanations of the methods are insufficient. What were the definitions of individual outcomes and effect measures? They should be clearly explained, at least, corresponding to the Results section.

We thank reviewer 3 for this important point, and we have included a section in the Methods to clarify our outcomes and effect measures.

“Exposure and Outcomes

In this study, we first investigated the causative organisms for infective endocarditis and compared this between patients who had recent dental procedures to those who did not. Infective endocarditis was diagnosed according to the Duke criteria. Dental procedures included tooth extraction, fillings, scaling, endodontal and periodontal treatment, implant insertion and root canal treatment. Causative organisms were identified by microbiology investigations. We then analysed the proportion of patients with IE who underwent dental procedures within 3 months prior to the diagnosis of IE. The proportion of patients with exposure to recent dental procedures were compared between patients with IE and those without IE. To understand the risk of IE post-dental procedure, the proportion of high-risk patients who developed IE within 3-months post-dental procedure was analysed. High-risk patients were defined based on the AHA/ESC guidelines (Table 1).”

2. The statistical analysis methods should also be explained in detail, at least, corresponding to the Results section.

The details of the statistical analysis methods used in this paper have now been added to the Methods section.

“Statistical tests were performed using the *meta* and *metafor* packages on R (R Core Team (2021)), based on the guidelines outlined in the Cochrane Handbook. Meta-analysis of IE and dental procedure proportions were performed using the random-effects inverse variance model with logit transformation and DerSimonian and Laird method, and subgroup analysis was performed based on the region of publication, year of publication and proportion of patients with antibiotic prophylaxis. The

type of bacterial organisms isolated from blood cultures were also analysed by meta-analysis of proportions as detailed above. Random-effects meta-analysis of binary outcomes was performed for case-control studies using the inverse variance model, where proportion of IE cases with exposure to dental procedures was compared to controls without IE. Sensitivity analysis using the random-effects generalised linear mixed models was also performed. Between-study heterogeneity was presented using the I^2 statistic. Publication bias was assessed by funnel plot and Egger's test for funnel plot asymmetry for analysis with more than 10 studies. A p value of <0.05 was considered statistically significant."

3. What study designs were considered in the study search? It should be explicitly mentioned. (study designs should be concretely mentioned)

We thank the reviewer for this suggestion, and the study designs included and excluded have now been added to the Methods section.

"Randomized controlled trials, cohort studies, case-control studies, cross-sectional studies and case series with 10 or more cases were included. Conference abstracts, reviews, case reports and case series involving fewer than 10 cases and non-English studies were excluded."

4. How the risk of bias was evaluated?

The quality of the studies included in the systematic review was assessed using the Newcastle-Ottawa Scale. The results are shown in Supplementary Table 1.

"Quality assessment"

The quality of the studies included in the systematic review was assessed using the Newcastle Ottawa Scale for cohort studies, case-control studies and case series."

5. How the proportion-type outcomes were defined? Several study designs were mixed, and I guess the estimation might be difficult for the case-control-type studies (especially if matching was used). The definitions should be clearly shown for individual study designs.

We thank the reviewer for this comment. For all included studies, proportion of IE patients who underwent recent dental procedures were estimated only in the cases population of IE patients. Therefore, in case-control studies, data on control patients without IE were not included in the analysis. Similarly, for the proportion of patients who underwent dental procedures who develop IE, only patients who underwent dental procedures were included, while controls who did not were excluded. In the proportion analysis for causative organisms, only IE patients were included by definition. Case-control studies were also analysed separately in Figure 2. This has been clarified in the Methods section.

“Meta-analysis of IE and dental procedure proportions were performed using the random-effects inverse variance model with logit transformation and DerSimonian and Laird method, and subgroup analysis was performed based on the region of publication, year of publication and proportion of patients with antibiotic prophylaxis. The type of bacterial organisms isolated from blood cultures were also analysed by meta-analysis of proportions as detailed above. Random-effects meta-analysis of binary outcomes was performed for case-control studies using the inverse variance model, where proportion of IE cases with exposure to dental procedures was compared to controls without IE. Sensitivity analysis using the random-effects generalised linear mixed models was also performed. Between-study heterogeneity was presented using the I^2 statistic. Publication bias was assessed by funnel plot and Egger’s test for funnel plot asymmetry for analysis with more than 10 studies. A p value of <0.05 was considered statistically significant.”

6. This study is a systematic review based on non-experimental studies. This is a relevant information, thus should be noted in Abstract.

We thank the reviewer for this suggestion. We have clarified that only observational studies were included in the Abstract.

“Of the 3771 articles initially identified and screened, 38 observational studies fit the inclusion criteria and were included in the study for subsequent analyses.”

7. The statistical software and packages used in data analyses should be noted.

This information has been added to the Methods section.

“Statistical tests were performed using the *meta* and *metafor* packages on R (R Core Team (2021)), based on the guidelines outlined in the Cochrane Handbook.”

8. What methods were used in the meta-analyses of proportion-type outcomes? (logit-transformation?) They should be accurately explained.

We thank the reviewer for this suggestion, this information has been added to the Methods section.

“Meta-analysis of IE and dental procedure proportions were performed using the random-effects inverse variance model with logit transformation and DerSimonian and Laird method, and subgroup analysis was performed based on the region of publication, year of publication and proportion of patients with antibiotic prophylaxis. The type of bacterial organisms isolated from blood cultures were also analysed by meta-analysis of proportions as detailed above. Random-effects meta-analysis of binary outcomes was performed for case-control studies using the inverse variance model, where

proportion of IE cases with exposure to dental procedures was compared to controls without IE. Sensitivity analysis using the random-effects generalised linear mixed models was also performed. Between-study heterogeneity was presented using the I^2 statistic. Publication bias was assessed by funnel plot and Egger's test for funnel plot asymmetry for analysis with more than 10 studies. A p value of <0.05 was considered statistically significant."

9. The synthesis result in Figure 2 is quite unnatural. The confidence interval was [0, 1], although those of individual studies were much narrower. Why such result was obtained under $\tau=0$? I think this result is incorrect.

We thank the reviewer for highlighting this issue. As there was no event in all the studies included in Figure 2, it may be the case where the maximum-likelihood estimator method for τ^2 is not valid. Using the DerSimonian Laird estimator for τ^2 , the resultant 95%CI is 0.0017-0.0270. Figure 2 and the Results section has been updated to reflect this analysis.

10. This study is a systematic review of non-experimental studies, so confounding is a relevant concern. The authors should mention for the control of confounding factors in individual studies, and should discuss how it potentially influences to the overall result.

We thank the reviewer for this suggestion. We agree that the analysis where two arms were compared i.e. the meta-analysis of case-control studies or case-crossover studies comparing the prevalence of recent dental procedure in IE versus control patients, may be susceptible to confounding factors given their observational and non-randomized nature. However, this is minimised in the individual studies. Porat Ben-Amy et al. 2009, Chen et al. 2015, Tubiana et al. 2017, and Chen et al. 2018 performed case-crossover studies where each case is their own control, thus minimising baseline differences in clinical factors. The other case-control studies e.g. Strom et al. 1998 used age-, sex- and residence-matched controls, while Lacassin et al. 1995 matched the age, sex and underlying cardiac condition. For the meta-analysis of proportions, the presence of confounding factors may be less relevant but differences in clinical and study factors may explain the high inter-study heterogeneity, which we have explored using subgroup analysis and metaregression analysis. We have discussed this in greater detail in the Results and Limitations section.

"Four studies were case-crossover studies, where cases were their own controls, thus minimising differences in baseline characteristics, while the other two studies used matched control populations

to reduce confounding factors. Strom et al. used age-, sex- and residence-matched controls (25), while Lacassin et al. matched the age, sex and underlying cardiac conditions between cases and controls (26).”

“In the comparison of IE patients with controls, case-control studies are susceptible to confounding factors, thus causality between dental procedures and IE cannot be definitively established.”

11. The I² statistics were quite large for many outcomes, even for subgroup analyses. It might be caused by the differences of background factors of individual studies. I recommend to perform meta-regression analyses for primary outcomes and relevant factors. Also, the causes of heterogeneity should be discussed in the Discussion section.

This is a valid point, and we have also highlighted the high heterogeneity between studies in the Limitations section. To investigate this, we performed the subgroup analysis using variables with sufficient data. We have performed additional meta-regression analysis using antibiotic use and year as continuous factors, which did not reveal significant relationships with effect size (proportion of IE patients who underwent recent dental procedures). Due to limitations in the data available from primary studies, additional meta-regression with other variables could not be performed. The meta-regression bubble plots are added to the supplementary materials and described in the Results section.

“Meta-regression for percentage of antibiotic use (intercept co-efficient -0.809; p=0.653) and year of publication (intercept co-efficient -0.005; p=0.772) revealed no significant relationship with the proportion of IE patients who underwent recent dental procedures.”

Metaregression for antibiotic use

Intercept coefficient: -0.8091, p=0.6529

Metaregression for year of publication

Intercept coefficient = -0.005, p=0.7717

12. Publication bias evaluations are recommended for primary outcomes.

We thank the reviewer for this important suggestion. We have assessed publication bias for meta-analysis with 10 or more studies. For the outcome of proportion of IE patients who underwent recent dental procedures, the funnel plot shows asymmetry, confirmed on Egger's test ($t = -4.46$, $p=0.0001$). The meta-analysis of the proportion of Staphylococcus and Enterococcus IE showed asymmetrical funnel plots while the proportions of Streptococcus, Streptococcus viridans, HACEK and culture negative IE did not show asymmetrical funnel plots. The funnel plots are added to the supplementary materials and described in the Results section. The possible publication bias is also discussed in the Limitations of this study.

"Publication bias was assessed by funnel plot and Egger's test for funnel plot asymmetry for analysis with more than 10 studies."

"Publication bias

For the outcome of proportion of IE patients who underwent recent dental procedures, the funnel plot showed asymmetry, confirmed on Egger's test ($t = -4.46$, $p=0.0001$). The meta-analysis of the proportion of Staphylococcus and Enterococcus IE showed asymmetrical funnel plots while the proportions of Streptococcus, Streptococcus viridans, HACEK and culture negative IE did not show asymmetrical funnel plots. Therefore, there may potentially be publication bias and small study effects, and further research is needed to confirm the findings of this study."

Funnel plot for proportion of IE patients who underwent recent dental procedures

Test result: $t = -4.46$, $df = 26$, $p\text{-value} = 0.0001$

Funnel plot for proportion of Streptococcus

Test result: $t = 1.36$, $df = 14$, $p\text{-value} = 0.1968$

Funnel plot for proportion of Streptococcus Viridans

Test result: $t = 1.35$, $df = 21$, $p\text{-value} = 0.1921$

Funnel plot for proportion of Straphylococcus

Test result: $t = -3.19$, $df = 23$, $p\text{-value} = 0.0040$

Funnel plot for proportion of Enterococcus

Test result: $t = -2.40$, $df = 19$, $p\text{-value} = 0.0268$

Funnel plot for proportion of HACEK

Test result: $t = -0.23$, $df = 22$, $p\text{-value} = 0.8239$

Funnel plot for proportion of culture negative

Test result: $t = -1.43$, $df = 21$, $p\text{-value} = 0.1673$

VERSION 2 – REVIEW

REVIEWER	Thornhill, M. University of Sheffield, School of Clinical Dentistry Other than being a researcher with expertise and publications related to this subject, I have no competing interests
REVIEW RETURNED	10-Aug-2023
GENERAL COMMENTS	thank the authors. The changes they have made have significantly improved the paper
REVIEWER	Noma, Hisashi The Institute of Statistical Mathematics, Department of Data Science
REVIEW RETURNED	08-Aug-2023
GENERAL COMMENTS	All of my previous comments are adequately addressed. Thank you.